# Introducing the EPP house (topological space) method to solve MRP problems

**Balázs Gyenge**[1], **László Kasza**[1], **László Vasa**[2]*

**1** Faculty of Economy and Social Sciences, Operation Management and Logistics, Supply Chain Management, Marketing and Tourism Institute, Szent István University Hungary, Gödöllő, Hungary, **2** Management Campus, Széchenyi István University, Hungary, Győr, Hungary

* laszlo.vasa@ifat.hu

**Data Availability Statement:** All relevant data are within the manuscript, as all data used are calculated as part of the study model.

**Funding:** The author(s) received no specific funding for this work.

## Abstract

The problem of product and process planning analysed so far is how we can take advantage of our strategy in planning. Among the principles of manufacturing and service management concepts is that after planning demand, planning transformation is one of the key steps of integrated efficiency; it makes it possible to save costs that are not value adding and are not necessary from the customer's point of view. Currently, the methods of material requirements and capacity planning can be seen as classic solutions that are based on dependency relations between different resources, which can be dynamic in space and time. Measuring and recording capacities raise several problems in addition to the fact that our planning methods are not always satisfying. In the literature, the methods of material requirements planning or manufacturing resource planning (MRP) are not typically optimization methods, so they do not guarantee the best solution, and even if our planning methods were satisfying, several manufacturing restrictions (the time allowed, the complexity of the planning process, the lack of testing opportunities, etc.) could prevent us from reaching satisfying application. It is necessary to create a simple planning algorithm that can give the planner a greater degree of freedom and that would be simple and algorithmic in order to maintain continuous conscious control, putting an end to planning uncertainty and leading us to the best solution under the given conditions. The aim of our research is to introduce a novel, simple planning algorithm, similar to heuristic methods that eliminates the problem of defining the order quantity when applying traditional methods, which prevents us from determining in advance which method is desirable (causing unnecessary planning steps); computer-based solutions hide the causal relations of the methodology from the planner (causing unreliability uncertainty).

## 1. Introduction and definition of the problem

Material requirements planning or manufacturing resource planning (MRP) plays a key role in production planning, generating outputs—such as material supply requirements, production orders, capacity requirements, and their precisely programmed schedules—and using customer orders, bills of materials (BOMs), and product structures. With the powerful help of MRP, we can obtain precise inventory status records or planning schedules. Classical MRP

**Competing interests:** The authors have declared that no competing interests exist.

methods consider only the forward material flow, deal with the initial inventory state, and supply outputs such as the minimum order quantity (MOQ) at the upper limit for inventory and discrete supplying times such as lead times or cycle times. Many papers that include details about the MRP process can be found in the literature, and there are a wide variety of different solutions with different main issues to address. Although these models have been found by researchers to work well under certain conditions and some of them have found some acceptance in industry, the most sophisticated planning models are not widely used. Although most MRP systems are computerised, most of the programs have serious limitations, and they use straightforward procedures that are too simple and procedures that are too rigid, without any human control by hand. In the literature, the methods of MRP are not typically optimization methods at all, so they do not guarantee the best solution, and even if our planning methods were satisfying, several manufacturing restrictions (the time allowed, the complexity of the planning process, the lack of testing opportunity, etc.) would make us unable to reach satisfying application. It is necessary to create a simple planning algorithm that provides greater flexibility and a greater degree of freedom for the planner.

## 2. Literature review

If the manufacturing system (or the infrastructure operating it) has been selected, planned, and created, the process itself should be planned regularly as well. This is nothing but planning the manufacturing processes.

Planning a manufacturing process affects competitive features that are important in satisfying needs such as flexibility, availability, and lead time, so its strategic role is critical.

In practice, a number of problems have to be solved, such as the determination of the material requirements, the quantity and scheduled distribution (programming) of capacity planning, the determination of lot sizes, the management of upper and lower constraints, cost calculation and optimization, closed-loop design, reuse and remanufacturing (recycling), uncertainty and fuzzy characteristics. The measuring and recording capacity raise many problems, and the design methods in many cases are not satisfactory [1–3]. Static and dynamic discrete solutions have been proposed, but there are also stochastic solutions [4] with a number of constraints to deal with uncertainties. Several researchers have used fuzzy methods for mathematical modelling [4–8]. Serna et al. proposed parametric linear programming [9]. Yazıcı et al. proposed mixed-integer linear programming (MILP) for the problem of reverse material flow [10,11]. First, they proposed a nonlinear mathematical model with some breakdown function definitions, and then they used step-by-step linearization to transform it into a linear model. Recently, MRP models have tended to seek solutions using cost-based integer programming models [12].

Therefore, process planning belongs to one of the most complex decision-making areas of operations management, and we cannot state that we will always have satisfactory simple and general applications or models [13–18]. The goal is to support a manufacturing service decision, and consequently, the organization will be able to use its resources in a more efficient way "[that]. . .allows plans to be checked against the capacity to determine whether they are realistic and achievable" [16].

The operations managers' task is to have capacity, new supplies, and data suitable for the most important resources or outputs regarding planning, to know the future demands (orders) at a certain level as well as the costs of holding and ordering, and to build relations between individual components.

There are three main directions of the concept of material requirements planning that are totally different from each other, and they provide fundamentally different answers to the

basic question (what to manufacture, when, and in what quantity). They are MRP planning, just-in-time (JIT) planning, and the theory of constraints (TOC) [19]. The main goal of all these methods and concepts is to increase efficiency, to reduce stock demand, and to reach the best possible cost savings [20] while matching customer demand. Moreover, some researchers have emphasized the supplementary character of these methods in planning and control [13], and at the same time, there is agreement regarding their favourable effects on costs; however, their rates can vary [14]. "A numerical analysis reveals that average cost savings up to 25% are possible if the optimal policy is used instead of the benchmark approach" [20]. "A series of regression models . . .explain why some companies achieve more benefits from MRP than others", but ". . .there is no one overriding factor which guarantees MRP success" [21].

In our approach, the goal of the MRP methods is to forecast the expected demands of the built-in subparts as exactly as possible and to plan a cost-minimizing ordering and holding system. MRP is a method that exploits the logic of the dependent relations between the subparts while connecting to the specific elements, supplies and stocks of the technology to determine exactly when, how many and what materials are needed.

From the end products' consumption forecasts (determined by the so-called master production schedules), the material requirements of the component supplies can be determined in advance if there is an adequately coherent dependence logic or consistent system (see dependence logic and BOM) [14,17]. "The material requirements planning (MRP) . . .carries out calculations on a level-by-level basis down through a Bill of Materials, which converts the Master Production Schedule of finished products into suggested or planned orders for all the subassemblies, components, and raw materials" [17].

In contrast to the obvious benefits, the many restrictions of MRP systems' early rigid schemes often caused dissatisfaction among operations managers, whose attitude is similar today because of the MRP systems' drawbacks, which are as follows (see also [14]):

- Customer orders are hard to follow or cannot be followed at all, and orders cannot be pushed. Due to aggregation, this tracking problem reduces the customer service level, the set of information given to the customers, the flexibility points, and ultimately the services.

- Because of the rigid planning algorithm and the inflexible realization system, a safety stock must be established in planning, which affects the benefits achievable. Since the accuracy of planning is crucial, a fluctuation in supplies can cause larger problems in a system like this, and the level of the safety stock will be higher in order to address uncertainties. "Safety stocks in material Requirements Planning (MRP) systems are held as in any manufacturing system, to cater for uncertainty" [18].

- The method of calculating the lot size may vary, and the chosen methods and algorithms are fairly random, which puts strain on those who conduct and criticize planning, and this cannot be reduced. For example, "We also observe that the solution to the transportation problem . . .is affected by the lot-sizing rules employed" [22]. This means there is too great a responsibility for operations managers since they can never be certain whether their procedures, methods, and decisions regarding which option to choose are good; this can be solved in a "dogmatic" way by assuming that "our system is the best", but this is not sufficiently scientific, and it either constitutes or becomes an obstacle to development. (There may be a better method, but neither time nor resources are usually available to determine this).

In practice, these dilemmas can cause operations managers some uncertainties and headaches even in deterministic cases. Due to rigid solutions, uncertainties being treated as hard and exact information, rapid IT support and the development of estimation techniques, the contradictions mentioned above do not necessarily lead to large problems as in previous

times, which is why there is currently a renaissance of dependence planning logic; moreover, its appearance in complex systems matches production philosophies (such as in [19] and the idea that we should ". . .identify controls that reduce the risks of these inaccuracies" [23]). As systems become more complex and integrated, and due to the growing need for many aspects of interopreraility supply chain systems or sustainability needs of inverse logistics [24] a new problem arises and become increasingly important that the currently used applications (ERP systems, SAP, ORACLE accounting information systems [25–29]) do not allow different interventions in the well-prepared, but overly automated supply and material requirements plans. These plans are based on oversimplified logistics and supply automations, and that is the reason why we need to go back to the basics. Although most MRP systems are computerized, the MRP procedure itself is logical and can be performed manually, and we suppose that it is a problem that can be significantly simplified by heuristics. We will attempt this in what follows.

## 3. Materials and methods

### 3.1 The problem of quantity

As stated previously, material requirements planning may be influenced by other factors, which should be considered in addition to the technological supply time, the stocks available, and the aggregated demand after taking into account the effects of necessary materials and demands on one another.

The *minimum order quantity* is one of the most important of this type of factor, which can distort our new supplies and cause a temporary holding demand. In commerce, it may be said that the cause of minimum-order items is the suppliers' inflexibility; however, managers working on production planning know that in most cases, this is not due to suppliers. Instead, the procurers are the cause of the quantity, and their production cost-reducing concepts are at fault. These economic considerations are much more responsible for minimum order quantities than suppliers' flexibility deficiencies, although they have been greatly improved in recent decades. (For instance, different analytic solutions are known for determining the optimal quantity, such as economic order quantity (EOQ) models and other dynamic algorithms).

Whenever we have a real (net) demand, *we should make a decision on how much to actually order*. This is a quantity decision that mainly influences the result, the achievable efficiency, the relevant costs, etc. (Additionally, see [22]). By using a frequent approach, but with a reverse direction or logic, in many methodologies, determining the quantity is one of the most important causes of planning. In some cases, we can see the quantity as the starting point (capability), and we plan in terms of it (interpreted as a kind of restrictive factor). The only exceptions to this dialectical approach are the dynamic and/or evolutionary scenery searching algorithms that use more iterations, which can make searching for number variations possible. Commercial MRP software uses several techniques to determine quantity [13–18]. These include the following:

- 1. The lot-for-lot (L4L) technique for minimizing the average cost. This technique primarily represents planning without a minimum quantity, which gives a benchmark for the other versions. We can use the data of this technique to make a so-called aggregate plan, which can make determining a frequently used order quantity type possible. If frequent orders are economical and JIT material resource techniques are introduced, then the L4L method is very efficient. However, when the setup (production-starting) and new supply costs are high (meaning significantly higher than the holding costs) or plant management is incapable of putting JIT theory into practice, then the L4L method may be too expensive. The question is whether it is worth using the minimum order quantity and aggregating certain order periods

(cycles), and thereby saving some costs. Those applying the algorithm often have a solution, namely, being aware of the stocking and supplying costs, because they make a cumulative aggregate plan for every period and every order; then, for these alternatives, the rate of aggregation is considered adequate, and the average cost per unit is minimal. This principle matches the decision made based on economic rationality or economic criteria; however, the rate of aggregation (quantity) is not necessarily optimal, so it needs verification (a back test). The logic behind this method is that "if we maintain the average cost per unit, so the average order and holding cost at the lowest level based on the possible aggregations, then we can do well in the long run" is an appropriate principle, but due to the logic of the aggregation and the non-incremental logic of the system, this is either not definitely true or does not definitely give the best solution or one very close to it.

- 2. The economic order quantity or optimal order quantity (EOQ), which minimizes the total cost. This technique aims to apply the EOQ known in logistics. Two versions are widespread in practice: tabular and systematic methods. Using the EOQ method is recommended if the fluctuation of the gross demand that is to be satisfied is small. In the tabular solution, we should compare the accumulated stock costs and the accumulated order costs in such a way that in every row of the table, we aggregate one more order until we get to the last, which aggregates all orders. It is useful to choose the aggregation where the two costs are approximately equal (considering that the figures in the table are discrete and not continuous and that we prefer the lower-error-cost version). In the systematic solution, we use the Wilson formula familiar from logistics and its variants to determine the order size, where "D" is the average demand of the planning period, "S" (setup) represents the order cost, and "H" is the holding cost [13,14]:

- 3. The part periodic balancing (PPB) technique. This material resource order technique balances the setup and aggregated holding costs with a changing order quantity, meaning that in this method, the minimum order quantity is not constant. Applying this method is particularly economical if the variance of the order amount (gross demand) is large compared to the average. To apply this technique, the economic part period (EPP) value should be determined first, which is a ratio of holding and demand: EPP = D/H. The EPP shows how many units should be held throughout 1 period so that it has the same cost as 1 order. With the help of this ratio, we calculate the so-called cumulative net demand (the accumulated amount times the holding time), and if this value exceeds the EPP, then we restart the commutation logically. The order numbers of the groups aggregated in this way can be seen as lots that match the order fluctuations. This technique results in different quantities and different periods with cost savings.

- 4. The Wagner-Whitin Algorithm. Making use of the speed of computer-based calculations, this simulation method analyses every possible quantity in terms of costs and accepts as a solution the one that leads to the minimum cost. The Wagner-Whitin process is a dynamic programming model used to determine quantity, giving the calculations some complexity. It supposes a definite time horizon beyond which there are no new net demands. However, it gives good results despite this limit. This technique has rarely been used in practice, but with the development of production systems' service processes and information opportunities, it has become available. Ultimately, the end result of solutions such as this is very similar to that of the PPB method since various periods and various order quantities are used here.

For all of these methods except for the Wagner-Whitin algorithm, they do not guarantee an optimal outcome, only a satisfying solution at most, which should be checked by a back-test in every case, and it is generally accepted that there is always a better solution than the chosen

one. The Wagner-Whitin algorithm finds the best possible solution concerning cost criteria; however, its drawback is that it hides causal relations and computer algorithm errors, and the error costs of the alternatives close to the solution or similarly satisfying solutions are suboptimal in other terms. Another problem is that certain computer representations and commercial solutions are not consistent regarding the methods used, and due to simplicity or inadequate information, none of the methods mentioned above are used, but certain heuristics (unpublished) are applied (which do not truly guarantee the best solutions).

The question is still whether a simplified or other type of heuristic algorithm does or may exist that can reduce the number of combinatorial options in such a way that the optimal solution or a satisfying solution close to it can be determined without a computer.

## 3.2 Modelling problems

There are several problems of modelling that make both creating and applying models difficult. We summarize the uncertainty factors of the system as follows:

- *External effects* (external *disturbing factors*) are not often modelled, and these can significantly affect the end result to be modelled. In many cases, the common effect of these factors is to make the phenomenon to be modelled effectively chaotic.

- The *system model's reliability* is usually not sufficient because of the *data certainty* of information. Because of uncertain data, static, theoretical and practical (empirical) models often become useless for managers, so the confidence level of these methods decreases significantly.

- *Mathematical and modelling difficulties.* Although we currently have several types of modelling techniques (hyperbolic programming, convex-concave programming, quadratic programming, and stochastic programming models that take uncertainty values or utility functions into account) [30], the methods' feature that they are capable of handling a problem regarding only a certain factor and defining the target criterion is neither unambiguous nor consist from single element.

- *Some problems do not have linear relations.* Changes and consequences may *not occur gradually (incrementally)*. Systems (such as in the present problem) consisting of unmeasurable relations cannot be analysed, and their solution requires dynamic programming or more complicated scenario analyses than this or a simulation.

- *The modelling paradox* is the phenomenon that the more realistic a model is, the less well it can be optimized. The dilemma of modelling is that the model should be simplified enough so that we can work with it, and at the same time, it should be complex enough that we can take its results seriously.

- *The trap of decision-making* is that we either accept or reject the suggestion, namely, the result obtained by the optimization models, or, which is less likely, we use the "partial results". The optimization models do not pay sufficient attention to the suboptimal solutions, which are very close to the optimal one but do not reach it, although they are probably of the same importance for the decision-maker; they can be valuable regarding other factors since several relations that are not to be modelled can give the reason for the solution. A similar problem arises if the decision-maker cannot truly trust the result obtained due to the complexity of the methods applied (because of the "closedness" of the solutions algorithmized by a computer application), which is why he or she is not truly interested in its application. If applying the system is very static (due to system uncertainties), even the best (optimization) models cannot be satisfying for decision-makers.

Because of the factors reviewed above, the information technology tool system of decision support is widespread and various, although its application is often far from expectations. In this study, we look for a method that saves more cost than the methods in the literature (preferably in an optimal way); additionally, the decision-maker should be able to follow and control the solution while avoiding "the trap of decision-making" described earlier. Our goal is *to increase the decision-maker's activity level (association plan)*.

## 4. Results and discussion

It is generally known that companies' results depend on the combination of resources, the strategy chosen, and a multitude of scientific approaches, which also implies that the processes can be predicted and optimized by a thorough analysis of the systems and a better understanding of the relations. Therefore, to *develop the MRP problem's solution*, we recommend *the topological solution as follows*:

Defining the problem:

Suppose that the demand necessary in an interval consisting of n periods is given. We would like to satisfy this demand with a minimal cost so that there are two types of costs: the unit cost of the order, which is constant in the case of an amount procured randomly, and the holding cost, which is available for a unit demand at the end of each holding period. In our study, we look for a method with which we can find the minimum total cost of the given interval or approximate it in an appropriate way.

Example: There is a property where cattle are raised. A river runs through the property. The cattle pasture is on one side of the river, and the slaughterhouse is on the other side. The cattle that are intended to be sent to the slaughterhouse are transported by ferry to the pasture where they can graze for free. The cost of transporting by ferry is 300 liards (quasi-order cost). However, the cattle waiting at the slaughterhouse should be fed in the evening, meaning a cost of 10 liards per head of cattle (matching the holding or storing cost).

If the weekly demand is given for each day, on which day should cattle be transported by ferry to the fold, and how many should be transported, to achieve the minimum cost?

### 4.1. Terms needed for the solution

- *Starting stock*:

Is the quantity of the starting stock important for us? Is it possible to transport the cattle back to the pasture? Concerning our exercise, let us suppose that there is a starting stock "$K_0$" that is in the fold at the moment of our study. We suppose that the cattle cannot be sent back (the order that came in should be served), and we should feed them until they go to the slaughterhouse (meaning that we should hold the surplus). We represent the decrease in the starting stock in the table below (Table 1):

The table above shows that in the case of demand $D_1 \ldots D_n$ ("D"), the starting stock runs out in period $P_n$. Until the day this happens, there are no orders or order costs, only holding costs, so our total cost is determined.

**Table 1. Matching the starting stock with 0 starting stock.**

| Period | | $P_1$ | $P_2$ | . . . | $P_n$ |
|---|---|---|---|---|---|
| Demand (e.g., the number of cattle): | | $D_1$ | $D_2$ | | $D_n$ |
| Stock: | $K_0$ | $K_0$-$D_1$ *where* $K_1$: = $K_0$-$D_1$ | $K_0$-$D_1$-$D_2$ *where* $K_2$: = $K_0$-$D_1$-$D_2$ | | $K_0$-$D_1$-$D_2$-. . .-$D_{n-1}$ = a *where* $(D_n$-a = $D_1^{'} | K_0 = 0)$ |

On the day when we run out of our starting stock (in the period $P_n$, where $D_n \geq K_n$), we use only the amount "a" without order, meaning that at this point, we should start a new time horizon in which we cannot order. The first period of this can be $P_n$, with demand $D_n$-a and stock K = 0. If $D_n$−a = 0, this period can be skipped concerning the order, and we can jump to the next relevant period; if $D_n$−a>0, then the demand $D_n$-a can be seen as the first demand $D_1^{'}$ of the next period. In other words, any case where there is a starting stock can be matched with another case where our starting stock is zero, namely, $K_n$ = 0. This is why we plan with the value $K_0$ = 0, which equals the closing stock of a theoretical previous period.

- *Closing stock ("$K_n$")*:

  Every stock in the table is the closing stock of a given period, called stock for short.

- *Demand ("$D_n$")*:

  The quantity (gross demand) that should be satisfied in a given period.

- *The meaning of the EPP number*:

  In the example above, the holding unit cost is H = 10, while the supply or order cost is S = 300. According to the definition, the relative proportion of the two costs EPP = S/H shows the maximum amount of reasonable holding, since a larger holding than this can be organized more cheaply by a procurement or an order. In our case, this value is 30 holding units, which means that it is not worth holding more than 30 units for a period of time. If the cost of holding is a unit, then we always order the amount matching the demand (Table 2).
  Concerning the simplification above and the demands, we have the following theorems:

*Theorem 1*: In the case of the demand $D_1$ in period $P_1$, quantity $R_1$ ("R") is ordered. The order arrives in the same period and satisfies the demand since there is no starting stock, so $K_0$ = 0; if there is a starting stock, we can write an equivalent problem in which there is not one. For zero starting stock, the demand should be secured in total, so demand $D_1$ will be ordered.

*Theorem 2*: $K_n$ = 0. By the end of the last day of the analysed period, the stock always decreases to 0 since we are looking for the particular solution where the cost is minimal; if $K_n$>0, then holding the stock earlier or later surely incurs excess cost, which is not acceptable.

*Theorem 3*: $R_m = \Sigma D_{m...i}$. In the case of the first order in the period chosen randomly, the demand of the first period must be procured, but further demands may be procured.

*Theorem 4*: $R_m = \Sigma D_{m...i}$; $K_{m-1}$ = 0 in the case of $R_m$, meaning that it is true for the "m"-th order that the stock of the previous period ran out (or the previous closing stock is zero, regardless of how many periods we can satisfy with the earlier order or how many we will satisfy with the next one). This theorem is equivalent to János Benkő's (2014) [31] conclusion that "the rule of running out" verified in the case of models tracking nonsteady (stock) demand, periodical stock, is solved by recursive dynamic programming; this rule states that "those varieties are always cheaper when we fill up the stock only after running out totally, so there is no refill". After this rule is combined with the theorem of sequential solution, the

**Table 2. Demand and orders.**

| Period | $P_1$ | $P_2$ | $P_3$ | $P_4$ | $P_5$ | $P_6$ | $P_7$ |
|---|---|---|---|---|---|---|---|
| **Demand: $D_{1...n}$:** | 15 | 20 | 6 | 10 | 8 | 20 | 5 |
| **Order: $R_{1...n}$:** | 15 | 20 | 6 | 10 | 8 | 20 | 5 |

optimal policies must consist of optimal subpolicies, which is why the optimum of the sub-periods from the time of filling up to running out is part of the optimal distribution of the whole period as well. Moreover, see [20]: "The optimal (s, S) policy is based on dynamic programming". We affirm that Theorem 4 can be derived by combining Theorems 1 and 2 if we put the limits of the aggregated period somewhere else again and again dynamically, and we always reuse the first two theorems.

- *Total cost ("C")*:

In the fundamental example given above, the holding unit cost is H = 10, and the order cost is S = 300. Calculating the total cost (or simply the cost) of a certain distribution (a certain alternative) can be defined as follows (Eq 1):

$$C = H \cdot \sum K_{1\ldots n} + S \cdot m \tag{1}$$

The total cost comprises the sum of the accumulated cost of holding the closing stock in individual periods and the product of the number of orders needed and the order cost. We note that the stocking cost is calculated in the period when it is held for at least one period (not in the period of its arrival, and therefore, the problem analysed follows a certain kind of logistics view). However, we find that as far as solving the fundamental allocation problem is concerned here, the "real costs" of holding and the total cost are almost irrelevant (from the view point of the method); they do not influence the possible methods of solving the analysed problem, just the specific mathematical expressions. The relation described above can also be traced back to the unit holding cost in the following way by dividing by H (see Eq 2):

$$C/H = \sum K_{1\ldots n} + S/H \cdot m \qquad \text{or} \qquad C/H = \sum K_{1\ldots n} + EPP \cdot m, \tag{2}$$

which can also be used to determine the minimum total cost.

- *Order ("$R_n$")*:

The amount by which the stock increases, which is used to satisfy the demand. The orders are denoted by $R_n^i$, where "n" is the period index of the order arriving and "i" is the number of periods covered by one order.

*Theorem 5*: If $R_m = \sum D_{m\ldots I}$, we order by increasing the order in the integer amount of the demand; thus, the whole amount ordered will be used by the end of the aggregated period. This rule is reasonable, meaning that if the surplus held is less than the sum of the demands so far and the next one, then we should have an additional order in the next period, which may cause extra costs. This will not serve our aims since the order cost does not depend on the quantity procured. In summary, we order in the following way: the amount of the order may or have to (or should) be increased by the integer sum of the next periods. The real question is how many of the next numbers should be aggregated concerning the order. (See Eq 3).

$$R_i^k = \sum\nolimits_{j=1}^{(k-1)} (D_i + D_{i+j}, \ldots D_j) \tag{3}$$

In our model, we look for the *order chain* (for all possible variations, namely, $2^{n-1}$ order chains altogether) for which the total cost is minimal. For the order chain (or the series of orders that cover the whole period, although each element can be an order covering several periods), it is true that $R_1^{i1}, R_{i+1}^{i2}, \ldots, R_{n-(im-1)}^{im}$ is an order series in which $R_n \neq 0$, the first order is $R_1^{i1}$, and the last order is $R_{n-(im-1)}^{im}$. There is no restriction on the number of elements

in the order chain (which may consist of only 1 element), and in the expression above, the number of elements is denoted by "m".

## 4.2. General combinatorial solution

In the case of a randomly given series $D_1 \ldots D_n$, the method of finite tries (the so-called computerised brute-force method) is available for us, where we calculate the sum of the order costs as well as the holding costs for each order and holding combination possibility; then, we can say of the least cost that we can find the optimum or the allocation with the minimal cost. The development of information technology has made applying the brute force method possible. Our goal was not only to find the best possible solution but also to construct the decision-maker's conscious control so that it would pay attention to the aspects that cannot be formalised. To widen our point of view, the methodology of writing all variations becomes important for us. Therefore, concerning the programming technique, we are searching for a width graph in which we traverse all branches.

During the visualization of the analysed alternatives, each solution can be matched with a series of numbers in such a way that the sum of the number series is "n" and the numbers show how many "P" periods the different "R" orders cover.

*Example 1*: 1111111. This number code describes the combination in which every period during an interval of seven periods gives the number of orders necessary only for the given period, meaning that only one period is served (see above). (We introduce a simplification to write the combinatorial possibilities, so we write the number of periods for which the given order is sufficient instead of the specific amount of the order).

*Example 2*: 241. This code describes the solution in which the order $R_1$ satisfies demands $D_1$ and $D_2$ in periods $P_1$ and $P_2$ (35 in the example). Then, the second order $R_2$ is enough to satisfy the demands $D_i$ for the next four periods $P_3$, $P_4$, $P_5$, and $P_6$, and finally, the last order $R_3$ is for the remaining period, namely, $P_7$. In short, $R_1{}^2$-$R_3{}^4$-$R_7{}^1$.

If n = 7, according to the binomial theorem, we should write n-1 for a total of $2^6$ = 64 different cases and calculate the total cost to find the minimum. (See Table 3).

**Table 3. Combinatorial map.**

| | | | | |
|---|---|---|---|---|
| *1* | 1111111 | 11221 | 1132 | 142 |
| *2* | 111112 | 12121 | 1312 | 412 |
| *3* | 111121 | 21121 | 3112 | 421 |
| *4* | 111211 | 12211 | 1321 | 133 |
| *5* | 112111 | 21211 | 3121 | 313 |
| *6* | 121111 | 22111 | 3211 | 331 |
| *7* | 211111 | 1114 | 1222 | 223 |
| *8* | 11113 | 1141 | 2122 | 232 |
| *9* | 11131 | 1411 | 2212 | **322** |
| *10* | 11311 | 4111 | 2221 | 16 |
| *11* | 13111 | 1123 | 115 | 61 |
| *12* | 31111 | 1213 | 151 | 25 |
| *13* | 11122 | 2113 | 511 | 52 |
| *14* | 11212 | 1231 | 124 | 34 |
| *15* | 12112 | 2131 | 214 | 43 |
| *16* | 21112 | 2311 | 241 | 7 |

**Table 4. Combinatorial cost map.**

| | | | | |
|---|---|---|---|---|
| *1* | 7x30 = 210 | 5x30+30 = 180 | 4x30+31 = 151 | 3x30+50 = 140 |
| *2* | 6x30+5 = 185 | 5x30+26 = 176 | 4x30+31 = 151 | 3x30+67 = 157 |
| *3* | 6x30+20 = 200 | 5x30+40 = 190 | 4x30+37 = 157 | 3x30+82 = 172 |
| *4* | 6x30+8 = 188 | 5x30+14 = 164 | 4x30+46 = 166 | 3x30+56 = 146 |
| *5* | 6x30+10 = 190 | 5x30+28 = 178 | 4x30+52 = 172 | 3x30+62 = 152 |
| *6* | 6x30+6 = 186 | 5x30+30 = 180 | 4x30+40 = 160 | 3x30+80 = 170 |
| *7* | 6x30+20 = 200 | 4x30+63 = 183 | 4x30+19 = 139 | 3x30+60 = 150 |
| *8* | 5x30+30 = 180 | 4x30+86 = 206 | 4x30+33 = 153 | 3x30+51 = 141 |
| *9* | 5x30+48 = 198 | 4x30+50 = 170 | 4x30+35 = 155 | **3x30+45 = 135 (min)** |
| *10* | 5x30+26 = 176 | 4x30+62 = 182 | 4x30+50 = 170 | 2x30+150 = 210 |
| *11* | 5x30+26 = 176 | 4x30+56 = 176 | 3x30+106 = 196 | 2x30+194 = 254 |
| *12* | 5x30+32 = 182 | 4x30+36 = 156 | 3x30+130 = 220 | 2x30+126 = 186 |
| *13* | 5x30+13 = 163 | 4x3+50 = 170 | 3x30+94 = 184 | 2x30+99 = 159 |
| *14* | 5x30+15 = 165 | 4x30+54 = 174 | 3x30+69 = 159 | 2x30+95 = 155 |
| *15* | 5x30+11 = 161 | 4x30+68 = 188 | 3x30+83 = 173 | 2x30+92 = 152 |
| *16* | 5x30+25 = 175 | 4x30+46 = 166 | 3x30+106 = 196 | 30+224 = 254 |

Based on the formula C/H = EPP·m+$\sum K_{1...n}$, we can determine the costs in the whole combinatorial map as shown below (the given values should be multiplied by "H" to obtain the total cost; see Table 4).

To calculate the cost, the amount of stock can be calculated from the demands in the individual periods to take the following equivalence into account: "...there is a stock series belonging to every order in the *order chain* for which the following is true" (Eq 4):

In the case of $R_i^j$, the total holding is

$$\sum (K_{(i+1)...(i+(j-1))}) = D_{i+1} \cdot 1 + D_{i+2} \cdot 2 + \ldots + D_{i+(j-1)} \cdot (j-1). \tag{4}$$

This means that the given numbers should be held as many times as the number of periods because they are far from the period in which the order arrives. We used this formula to calculate the holding demands in the table, to perform calculations for every element of the order chain and to add them. This table reveals that for the combination of the order chain $R_1^3$-$R_4^2$-$R_6^2$, the resulting minimum cost (we aggregate the first three periods concerning the order, then the next two and finally the last two) is 135 multiplied by the holding cost H = 10, which equals 1350 liards.

As seen above, in the case of finite tries for n = 7, we can obtain 64 cases; however, in the case of n = 21, there are more than 1 million possibilities, which can cause a problem. The literature calls the exponential increase of the alternatives a combinatorial explosion, which soon leads to the breakdown of procedural solutions.

However, this problem can be handled by recursive dynamic programming that proceeds backwards, as János Benkő (2014) [30] stated, increasing the number of periods in which the problem holds; we cannot see the possible solutions in front of us (only the so-called optimal solution or alternative solutions, and we are working with relatively many restrictions in finding the solution).

In what follows, we are looking for a method that eliminates fast "wrong" possibilities (those that are not favourable concerning the cost), which drastically decreases the number of necessary tries and therefore the size of the combinatorial space to be analysed. By using heuristic rules, our goal is to drastically reduce the set of possible combinations so that among the remaining potential solutions, the option with the minimum cost can still be found easily (can

be followed by everyday logic), as well as the options that are close to this solution but not the same.

## 4.3. Describing and creating the EPP house

To follow the solution visually, we introduce *the term topological map*, which *helps in searching for a solution and analysing (interpreting) the algorithm applied by visualizing the coherent elements*. The EPP house is such a matrix, and its topological arrangement makes possible the combinatorial analysis of the related demands.

*Step 1*: We write the values $D_1 \ldots D_n$ in the lowest row of an nxn table, then write the values $D_2 \ldots D_n$ above $D_1$, as seen in the table, the values $D_3 \ldots D_n$ above $D_2$, and so on. Visualizing the logic of "going from back to the front", the setup or sketch of the EPP house enhances the overview of the combinations. Since we would like to avoid redundancies in the formalization of the problem, it is sufficient to draw only half of the table or the matrix, meaning only the parts on the left of the diagonal, because this will truly contain all possible combinations. (Table 5).

*Step 2*: Instead of the table format, we can apply a skew matrix below or even a triangle that can lead to a much more intuitive drawing as follows: its base is the demand from left to right, the left-hand side of the triangle from top to bottom is the reverse descending chain of demands, and its right-hand side is the repetition of the demands in a descending chain from back to front. The EPP house itself is a triangle that can be covered by $2^6$ small triangles (partial triangles) inside. (Fig 1).

Based on the illustrations, we can say that any order chain can be drawn in an EPP house or in a pyramid. In these house-like shapes, all 64 order chains written as a general combinatorial solution can be analysed visually, and hopefully, this forms the basis for further conclusions. According to the initial logic, every R order can be definitively assigned to individual periods (and their demands), and it can be determined when ordering how much holding cost will accumulate while we satisfy the demand of the next day. Based on our presuppositions (hypotheses) from the drawing in the EPP house, we can read all information necessary to determine the total cost.

From the example in Fig 2, we can see that the period of planning can be covered by two orders (the $R_1^4$–$R_5^3$ order chain where m = 2), which can be seen from the two partial triangle vertices ("mountains") coloured blue. As we move along the vertices of the partial triangles, we can read the following: the amount of the $R_1^4$ order equals 15+20+6+10 = 51, of which 15 is used immediately without holding while the rest is held; the second order will be $R_5^3$ = 8+20 +5 = 33 (84 altogether). It can also be seen while drawing the EPP house according to the horizontal and vertical accumulation of the areas coloured light blue that the total holding needs

**Table 5. The basic matrix (topological map) of the EPP house.**

| 5 | | | | | | |
|----|----|----|----|----|----|---|
| 20 | 5 | | | | | |
| 8 | 20 | 5 | | | | |
| 10 | 8 | 20 | 5 | | | |
| 6 | 10 | 8 | 20 | 5 | | |
| 20 | 6 | 10 | 8 | 20 | 5 | |
| 15 | 20 | 6 | 10 | 8 | 20 | 5 |

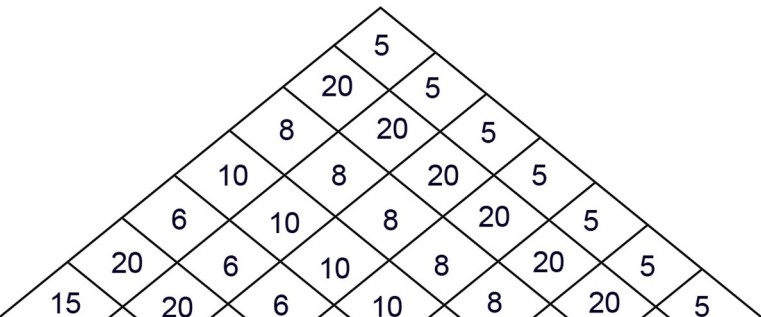

**Fig 1. EPP house or pyramid (topological map).**

are $K_1^4 = (K_1+K_2+K_3+K_4) = 36+16+10+0 = 62$ and $K_5^3 = (K_6+K_7+K_8) = 25+5 = 30$, adding up to $\Sigma K = 92$.

Thus, the (cumulative) demand of holding stock for the designated alternative $\Sigma K = 92$ (where $K_1 = 36$; $K_2 = 16$; $K_3 = 10$; $K_4 = 0$; $K_5 = 25$; $K_6 = 5$; $K_7 = 0$) is obvious regardless of the demand $D_i$ of the white areas. The expression $K_1^4$ is formed similarly to $R_1^4$ above and can also be called a *cumulative stock chain*, which can always denote a cumulative amount. For example, in the table, we can see the stock demand of 10 in column $D_4$ circled three times in the light blue area, meaning that in order $R_1^4$, we should hold three times after that; this appears automatically when adding up the cost $K_i$. The number 6 is seen in column $D_3$ twice, meaning that it is held twice. (Fig 2).

As seen above, the possible orders can be visualized, and the alternative cost can also be determined easily. The cost of the distribution above can be calculated from the sum of the accumulated stocking cost and the accumulated order cost according to the cost function defined earlier ($C/H = \Sigma K_{1...n} + EPP \cdot m$). The cost of the distribution above equals $(92+2\cdot30)\cdot 10 = 1520$ liards, which is the technical cost multiplied by H. (See Fig 3).

*Step 3*: Since our task is to reduce the set of possible solutions drastically to simplify the task to a problem that can be solved easily by traditional human logic, the next task is to exclude

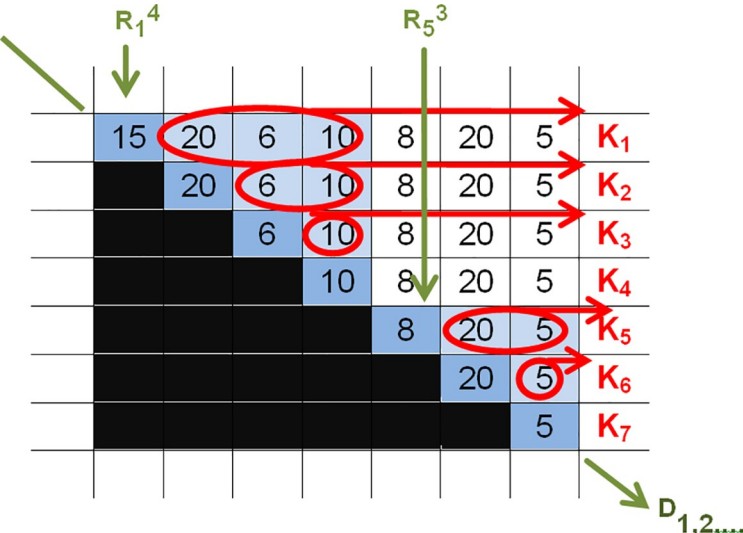

**Fig 2. The topological representation of an order chain of two orders.**

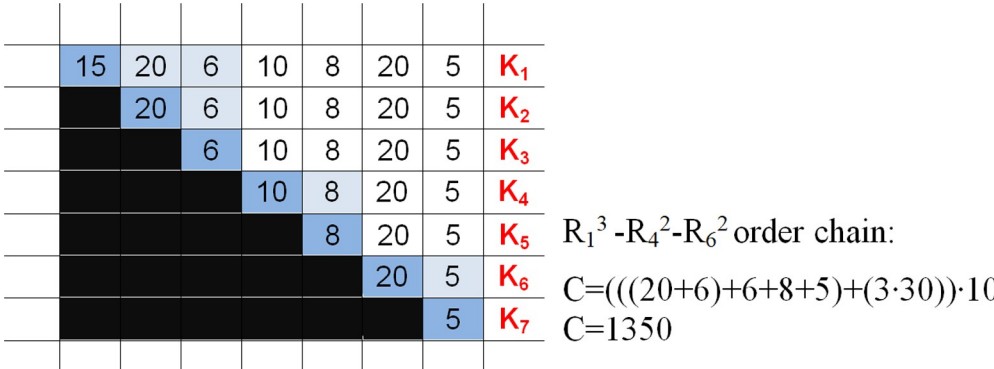

**Fig 3. Another alternative: Drawing the EPP house for the optimal distribution.**

the irrational solutions (orders) quickly by heuristic logic. Clearly, there are orders about which we can say that our order is not favourable, namely, that it is "irrational". It is obvious in our example that the order of the total demand of all 7 periods is absolutely irrational because we can find at least one resolution whose total cost is less than $7 \cdot 300 = 2100$.

Irrational premises:

A possible order is irrational if it can be resolved to two smaller orders that can cover the same period of time but whose total cost is less, such as $R_1^7$. Any further order is irrational if the sum of the stocks in the last column exceeds the EPP number, as seen in the chart above from its definition. (For instance, it is not worth holding 8 units for 4 days at a cost of 32 if the same amount can be supplied for 30 in the fifth period $D_5$). A good example of this is $R_1^5$, which is also irrational. Of course, aggregations larger than this are irrational as well. Based on a syllogism, it also holds true that every order chain in which orders termed irrational appear earlier is irrational as well, since the reverse principle of sequential optimization states that an optimal policy cannot contain a nonoptimal component (since its repair would lead to a better result). For example, $R_1^5$ can be resolved into a more favourable order chain, such as $R_1^4$-$R_5^1$ (so $R_1^4$-$R_5^1$-$R_6^2$ has a more favourable cost than $R_1^5$-$R_6^2$). (See Fig 4).

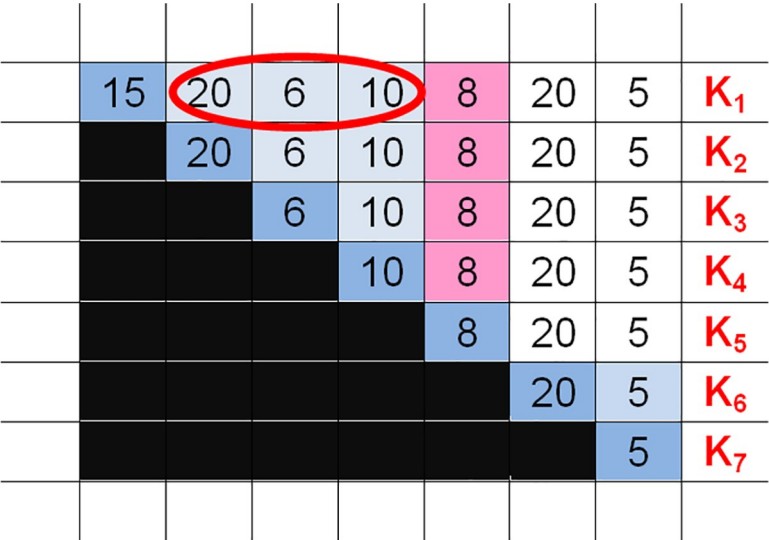

**Fig 4. Elimination and graphical representation of irrational orders.**

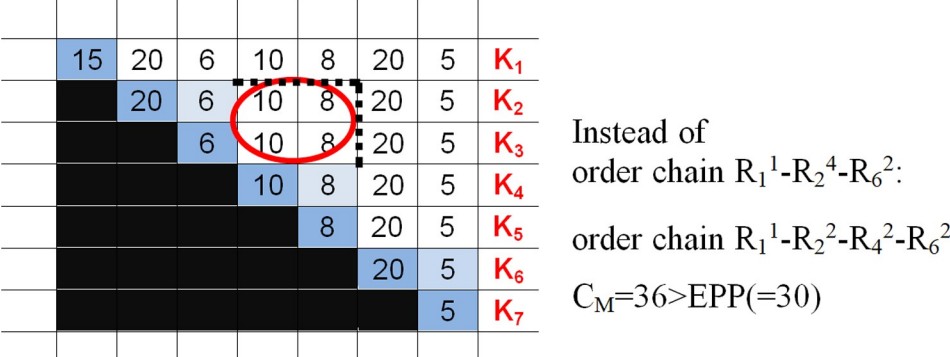

**Fig 5. Cost savings (and reasonability) of the EPP triangle resolution.**

Thus, we can observe that $R_1^5$ can be resolved (from the front or from the back), saving cost $C_M = 32$. Can $R_1^4$ be a part of the optimal strategy? The answer is surely no, since $K_1 = 36$ in the circled area (see Fig 4) is also greater than the EPP number, showing the reasonability of further resolution. Now, let us analyse the possible profit of resolving an aggregated order (triangle). For instance, if we resolve order $R_2^4$ in the middle, then either this area turns white (indicating the accumulated cost $C_M$ of holding free stock) or the profit is reasonable only if it is greater than EPP (for the order cost of the surplus, see the circled area in Fig 5); namely, $((10+8)+(10+8))>$EPP.

For every triangle, a rectangle of savings can be formed (possibly in several ways), and the cumulative amount of area covered is the savings, which should be greater than the EPP number. In the case of $R_x^i$ (i-1), rectangles (or resolution) can be determined, and the size of the rectangles is (i-p)xp, where p is a number greater than 0 but less than i. If an order (an arbitrary triangle) cannot be resolved economically in 2 smaller orders, this should be seen as a rational order forming a part of an alternative optimal resolution.

*Step 4*: Eliminating irrational (large or small) orders quickly. We can mark the irrational orders in the EPP house in such a way that we eliminate the vertices of the irrational order triangles ("hills"), and consequently, the given aggregation. We can do this easily because we mark a cell red if the cumulative sum in column $D_i$ below it is greater than the EPP number. Graphically, if any order chain drawn in the EPP house covers a red field, then that particular order chain is certainly irrational, so it is not necessary to calculate it (or to consider the related information technology) since it will not be a part of any rational alternative (nor of the optimal alternative). (See Fig 6).

According to Fig 6 above, the order chain $R_1^4$-$R_5^2$-$R_7^1$ should not be analysed since it is surely irrational. As we can see, only those combinations are worth analysing that are order chains going through the vertices in the thin lane between the black and the top-right red areas. A rational order chain consists of triangles ("hills") that totally cover the main diagonal, and then the vertices of the orders following one by one touch each other along the main diagonal. The "irrationally large" orders are order aggregations whose resolution can save costs, making it possible to eliminate "certainly sub-optimal" possibilities quickly. The lower the EPP is, the narrower the middle area where the vertices of the triangles of the rational combinations can be found. It is generally true that for every cell ("hill top") outside the main diagonal, we can find a value of savings (in a separate table) that can give every hill top all possible (i-1)

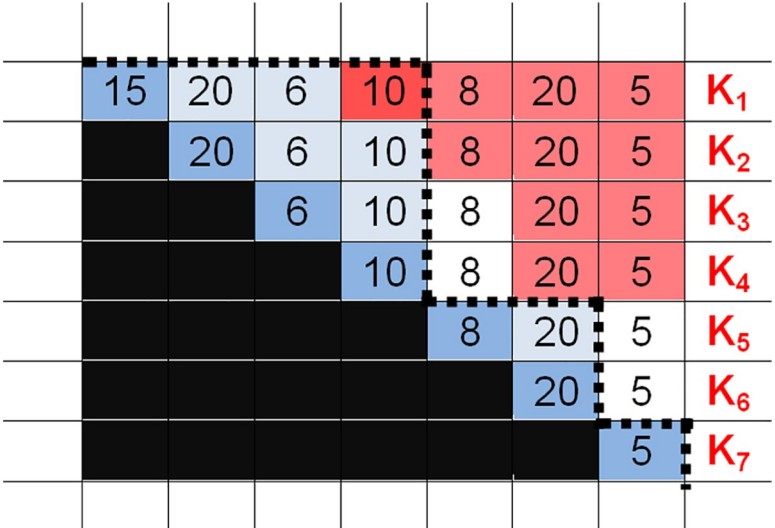

**Fig 6. The area of an irrational order in an EPP house.**

resolutions concerning the largest savings available. However, this helps in optimization, which is not our primary goal. In the separate table, all cells in which the values are greater than the EPP number are among the irrationally large orders and are marked in red, providing an alternative method of eliminating them quickly. Although, theoretically, there is a rational order chain whose cost is greater than that of a chain containing an irrationally large order, this does not influence the principle that these combinations do not have to be analysed.

After employing the elimination method for irrationally large orders, the question is whether there are "irrationally small" orders that can be eliminated. The answer is yes, since the 7 orders ($R_1^7$) in the main diagonal are of this type. An irrationally small order is defined as one with a holding cost surplus after aggregating two orders, meaning that the loss should be less than the EPP number. If an aggregation cannot bring an irrationally large order and the result is another rational order, then we can state with certainty that we are not analysing the optimal solution with the lowest cost. However, aggregation is available, and it is not necessarily needed because an alternative aggregation can yield a better chain. It is important to state that the optimal solution gives such a tight chain, in which aggregating any two elements can result in an irrationally large order. Unfortunately, we cannot determine the irrationally small order pairs in the EPP house, but we think that eliminating the irrationally small orders cannot give any benefit in the programming techniques since only by analysing them individually can we determine whether they contain elements to be aggregated. An even more important point is how Fig 6 is created. Through information technology support, we can obtain a visual tool in which, rearranging the vertices by "dragging" the mouse, we can easily compare the possible solutions, and we are able to observe how much the current solution differs from the value of the optimal (best) solution regarding the cost.

*Step 5*: Creating a design tool using Excel or any other platform. Thinking over these points, we can obtain a very strong and easy-to-use design tool if we create a visualization technique that shows a topological map as above and puts a handle on each "hill top" (vertex of a triangle) that can be moved with the mouse; these can break apart or merge due to any movement while we can continuously see the costs and the planned order chain, as shown below. (See Fig 7).

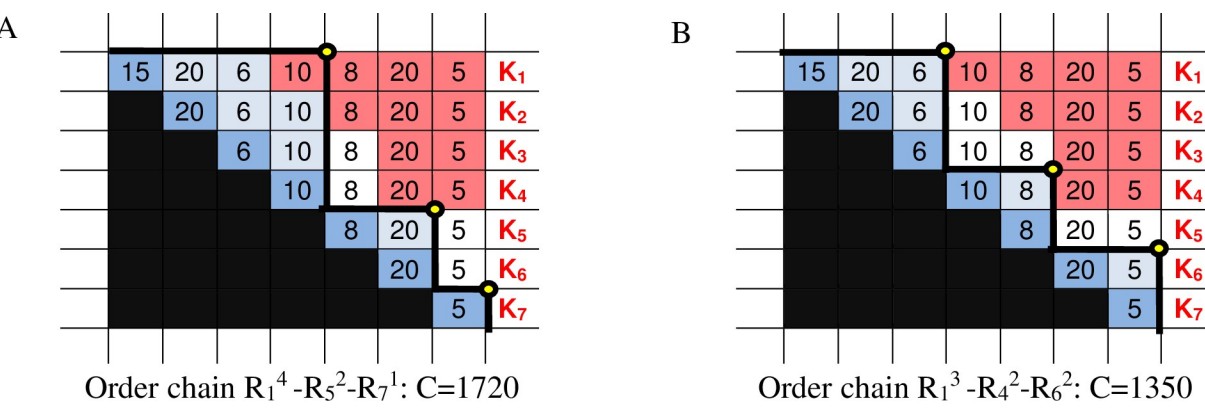

Order chain $R_1^4$-$R_5^2$-$R_7^1$: C=1720    Order chain $R_1^3$-$R_4^2$-$R_6^2$: C=1350

**Fig 7. Planning for the EPP house.**

In our research, it is worth planning the irrationally large orders' vertices starting from the top since we can easily obtain a "tight" order chain and, by aggregating any of its partial orders, obtain an irrationally large order, which is a special feature of its optimal or alternative variations. However, it is important to mention that operations managers performing practical planning are also interested in alternative rational (sub-optimal) solutions, for which we can imagine several factors that are unlikely to be formalised mathematically. For example, someone may prefer supplying on Friday to Monday or may want to avoid a particular day because there is a traffic obstacle due to a running race nearby, etc. Nonetheless, in practice, we would like to consider these fundamental or intuitive factors during our planning. This methodology gives an opportunity to mark these vertices (the potentially optimal order combinations) in a mathematically derivable way for the user, which can ensure the best exchange while making planning easier. In addition, these other factors can ultimately be observed during the process of interactive planning (e.g., in terms of certain capacities, customs, and supplier habits).

## 5. Conclusions

In our study, we deduced and proved that it is possible to make a procedure that is capable of reducing the number of solution options in such a way that it is easier to determine the optimal solution and its possible variations and to enable the manager to integrate his or her own intellect and his or her needs that are difficult or impossible to formalise mathematically into a solution process that operates visually and can be followed and controlled on an interactive surface; during its application, the decision-maker should feel comfortable. At this stage of our research, we have not yet developed an executable program or any kind of user interface. However, we can envision an interactive program that embeds any data from excel worksheets or EDI sources, and generates a large scrollable topological matrix on the screen, that highlights the path to critically suboptimal solutions using graphical anchors, handling nodes and colour management, with continuous recalculation of costs.

The advantage of this kind of solution is that you can see visually where the reasonable combinations are and how some changes affect costs. The manager does not have to favour a single "best" solution. So the final step depends on his satisfaction.

The MRP problem does not typically operate in a non-incremental way, and in this case, we cannot modify the results formalised by machine logic and mathematical analysis in small steps and adapt them to local needs and subjective aspects since a small change (order modification) can lead to a totally different cost error at an unpredictable rate. Experts respond by

building in expensive safety containers, or they do not accept the optimization results because these are too inflexible for them. In our opinion, the method described here can help address this problem.

The main limitation of the method is the size of the matrix, but I think we have shown that we may can handle up to 100 or more columns and rows at a time, or even more, which is more than enough in practice, because we very rarely plan more time periods in a cycle. (See theory of freezing in capacity planning). Using this technic, it provides to build in any user constraints and visualise them more interactively than before. This method gives as a quick easy-to-use scenario comparing technique.

In our analysis, we note that to create efficient order combinations, we do not necessarily have to know the specific order and holding costs, whose exact calculation is almost impossible in practice if we wish to determine it with scientific precision. It is sufficient to know only the ratio of the two factors in the form of the EPP number, and then the problem can be handled well with the technical costs. We see this statement as a radically novel approach.

The EPP house is not only a suggestion for solving a problem and a new methodology but also a certain way of thinking that draws attention to representing problems visually and integrating the decision-maker at a deeper level. We recommend our new topological method for those who find it difficult to integrate optimization results into everyday practice and want to create a higher degree of freedom for the decision-maker's intellect in their systems or processes. It is an interesting dilemma for us that in the world of artificial intelligence and big data in the 21st century, we almost move in a direction opposite to these trends by offering such a paradigm, in which we draw attention to the logic of the decision-maker and a natural way of thinking, and we feel that a greater level of this is necessary. This is imperative because there are problems and situations that can go beyond mathematical and machine logic and require the integration of human intellect. We present our solution procedure and concepts as a good start for further discussion and precedent.

## Author Contributions

**Conceptualization:** Balázs Gyenge, László Kasza.

**Data curation:** László Kasza.

**Formal analysis:** Balázs Gyenge.

**Investigation:** Balázs Gyenge, László Kasza.

**Methodology:** Balázs Gyenge.

**Project administration:** László Kasza.

**Resources:** Balázs Gyenge, László Vasa.

**Software:** László Kasza.

**Supervision:** László Vasa.

**Validation:** László Vasa.

**Visualization:** Balázs Gyenge, László Kasza.

**Writing – original draft:** László Kasza.

**Writing – review & editing:** László Vasa.

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
