## [Decision Letter · Decision Letter 0]

13 May 2021

PONE-D-21-09900

Introducing the EPP house (topological space) method to solve MRP problems

PLOS ONE

Dear Dr. VASA,

Thank you for submitting your manuscript to PLOS ONE. After careful consideration, we feel that it has merit but does not fully meet PLOS ONE’s publication criteria as it currently stands. Therefore, we invite you to submit a revised version of the manuscript that addresses the points raised during the review process.

We look forward to receiving your revised manuscript.

Kind regards,

Dragan Pamucar

Academic Editor

PLOS ONE

Journal Requirements:

5. We note you have included a table to which you do not refer in the text of your manuscript. Please ensure that you refer to Table 12 in your text; if accepted, production will need this reference to link the reader to the Table.

Reviewers' comments:

Reviewer's Responses to Questions

**Comments to the Author**

1. Is the manuscript technically sound, and do the data support the conclusions?

Reviewer #1: Yes

Reviewer #2: Yes

Reviewer #3: Yes

2. Has the statistical analysis been performed appropriately and rigorously? 

Reviewer #1: Yes

Reviewer #2: Yes

Reviewer #3: Yes

3. Have the authors made all data underlying the findings in their manuscript fully available?

Reviewer #1: Yes

Reviewer #2: Yes

Reviewer #3: Yes

4. Is the manuscript presented in an intelligible fashion and written in standard English?

Reviewer #1: Yes

Reviewer #2: Yes

Reviewer #3: Yes

5. Review Comments to the Author

Reviewer #1: The author/s have chosen an interesting topic that has been of constant interest over the past decades. MRP calculation is a constant topic among experts and its importance is constantly changing depending on the level of process management and the area of activity and process planning. Mathematically we have some methods to use, but none of them can be taken as an optimisation algorithm because its dynamic characteristics. This is why this article may be of interest to readers. Although, this “topological method” also not an optimisation, but it gives the user more freedom to compare possible solutions (or alternatives).

Please be sure that author is following the same format which is followed by the Journal. I would recommend to refer related literature published in Plos One and other important journals.

Authors synthesized literatures well and add their own opinions too, but I would suggest to improve the paper with more international sources. I appreciate the methodology developed, which based on a matrix logic but some more details would be useful such as the limitations of the model, or its use, and what other benefits are assumed.

They introduced their results in a well-structured way and summarized the advantages and I agree the conclusions of authors which they presented in the last chapter of the paper but I still don’t know how this method could be used in practice, as this method is highly interactive and not automated yet. I would like to ask you to interpret to me.

Reviewer #2: The paper analyses a significant evergreen topic, how to decide on a minimum order quantity without constant recalculation, or just adopt a kind of computerised solution automatically without any decision-making and management scope or influence. It is very actual and interesting topic: how to solve this problem with an alternative way. A challenging context indeed, however, the authors could manage the problem with a half heuristic method. From my point of view, I think this method cannot achieve the final state, how you can solve this problem? Please make a clearer statement about the closing step and the constraints of the result obtained and the restrictions on use.

The article contains an appropriate, critical and comprehensive literature review, however, I recommend extending it with some application based resources.

The logic and structure of the paper are acceptable; the findings are based on the outcomes from studied method and the conclusions are in coherence with the results.

Reviewer #3: Comments and Suggestions for Authors

Thank you for the opportunity to read the study paper and to find interesting issues concerning the topic debated. The subject is significant and relevant and the study contributes to the development of knowledge by proposing a new way of method, that offers the user more degrees of freedom and more controls over the use of different suboptimal solutions. Still, some clarifications are needed to improve the papers' value added to the literature, as suggested below.

Title: it is clear and expressive.

Introduction and literature review: the first impression of reading the introduction is that it is brief but informative and focusing on the narrowly defined topic.

In one of the authors' summary reflections, they write that " In our approach, the goal of the MRP methods is to forecast the expected demands of the built-in subparts as exactly as possible and to plan a cost-minimizing ordering and holding system." Please explain what you mean by cost-minimizing, as this is not necessarily required in MRP systems.

The literature review section is brief but informative. I suggest incorporating the newest debates of prestigious international journals.

Material and methods and the problem of quantity and modelling problems section, is more or less clear. Please clarify the following: The factors described are mainly of a technological and methodological nature. What about the soft skills and human factors such as barely formalizable and different aspects of collaboration.

Results and discussion

I advise the author to read the article carefully and re-examine their claims. Some statements, as the following one "However, we find that as far as solving the fundamental allocation problem is concerned here, the “real costs” of holding and the total cost are irrelevant" are very progressive without any arguments supporting it.

Explain what kind of restrictions you mean, when you write “there are many restrictions” … when you use backwards running, recursive dynamic programming method, “in finding the solution”

Conclusions

In the last section of the paper, the conclusion needs to include the paper theoretical contribution, practical/managerial contribution, limitations and future research perspectives.

Please add some more details such as the limitations, which would further enhance the professional quality of the article.

The paper has good potential but at this stage the paper further minor changes and some more clarifications is needed on the issues presented.

6. PLOS authors have the option to publish the peer review history of their article (what does this mean?). If published, this will include your full peer review and any attached files.

Reviewer #1: No

Reviewer #2: No

Reviewer #3: No

---

## [Author Response · Author response to Decision Letter 0]

25 May 2021

Dear reviewer ‘R1’

The authors have chosen an interesting topic that has been of constant interest over the past decades. MRP calculation is a constant topic among experts and its importance is constantly changing depending on the level of process management and the area of activity and process planning. Mathematically we have some methods to use, but none of them can be taken as an optimisation algorithm because its dynamic characteristics. This is why this article may be of interest to readers.

Thank you for valuable assessment. I would like to thank you for your valuable comments. 

Although, this “topological method” also not an optimisation, but it gives the user more freedom to compare possible solutions (or alternatives).

Yes. Our aim was to make a heuristic like methodology, which gives much more freedom for the user. I have added some supplementary words for it (see r26; r27). 

Authors synthesized literatures well and add their own opinions too, but I would suggest to improve the paper with more international sources.

Therefore I have made some changes in the paper (r136-143). I have added some new international sources (6 pieces: r136-143) mainly in the literature review section with a particular focus on application based issues. 

I appreciate the methodology developed, which based on a matrix logic but some more details would be useful such as the limitations of the model, or its use, and what other benefits are assumed.

I very much agree with all the suggestions and have corrected them in text. I highlighted the corrections with red colour in the text and in the reference section as well. (r638-643)

They introduced their results in a well-structured way and summarized the advantages and I agree the conclusions of authors which they presented in the last chapter of the paper but I still don’t know how this method could be used in practice, as this method is highly interactive and not automated yet. I would like to ask you to interpret to me. 

Thank you for valuable comments. I agree with the suggestion and have edited a new section in the conclusions with a brief explanation. I highlighted the corrections with red colour in the text. (r622-627)

R2

The paper analyses a significant evergreen topic, how to decide on a minimum order quantity without constant recalculation, or just adopt a kind of computerised solution automatically without any decision-making and management scope or influence. It is very actual and interesting topic: how to solve this problem with an alternative way. 

I would like to thank you very much for your valuable evaluation. 

A challenging context indeed, however, the authors could manage the problem with a half heuristic method. 

Our aim was to make a heuristic-like method that combines visuality with enumeration, which gives a heuristic-like method with a set of suboptimal solutions. (see r26; r27). 

From my point of view, I think this method cannot achieve the final state, how you can solve this problem? Please make a clearer statement about the closing step and the constraints of the result obtained and the restrictions on use.

I have explained it in the conclusions section. (see r627-629) I highlighted the corrections with red colour in the text. 

The article contains an appropriate, critical and comprehensive literature review, however, I recommend extending it with some application based resources.

I have added some new international sources (6 pieces: r136-143); I have expanded the literature review section with specific new resources, which related to applications, ERP and MRP relations, role of MRPs in accounting information systems, and new needs about supply chain and sustainability.

The logic and structure of the paper are acceptable; the findings are based on the outcomes from studied method and the conclusions are in coherence with the results. 

I agree with all the proposals and have corrected them in the text.

R3

Comments and Suggestions for Authors

Thank you for the opportunity to read the study paper and to find interesting issues concerning the topic debated. The subject is significant and relevant and the study contributes to the development of knowledge by proposing a new way of method, that offers the user more degrees of freedom and more controls over the use of different suboptimal solutions. Still, some clarifications are needed to improve the papers' value added to the literature, as suggested below.

Thank you for valuable and very precise assessment. I revised the text and I made some new clarifications and extra explanations. (see r26; r27); (r136-143); (r622-627); (r628-630) (r638-643) 

Title: it is clear and expressive. 

Introduction and literature review: the first impression of reading the introduction is that it is brief but informative and focusing on the narrowly defined topic.

Thank you for your comments on the introduction and literature review. 

In one of the authors' summary reflections, they write that " In our approach, the goal of the MRP methods is to forecast the expected demands of the built-in subparts as exactly as possible and to plan a cost-minimizing ordering and holding system." Please explain what you mean by cost-minimizing, as this is not necessarily required in MRP systems.

Technically yes, but usually we use MRPs and kaisen or any other decision support systems in process management to achieve faster cycle times, more balanced material flow, less inventory stock, and less costs so we don’t have to give up the original objectives, even if other constraints makes strategic conflicts. 

The literature review section is brief but informative. I suggest incorporating the newest debates of prestigious international journals.

I have included five new international resource (r136-143); (r727-743). These resources highlight recent discussions on the subject, and present the best-known solutions in the field of applications, ERP and MRP relations, role of MRPs in accounting information systems, and new needs about supply chain and sustainability.

Material and methods and the problem of quantity and modelling problems section, is more or less clear. Please clarify the following: The factors described are mainly of a technological and methodological nature. What about the soft skills and human factors such as barely formalizable and different aspects of collaboration. 

I believe that our new method offers a wider range of opportunities to use unique concepts in use and is incorporated into the model or decision support. (See explanation of the conclusions section. (r622-630) 

Results and discussion 

I advise the author to read the article carefully and re-examine their claims. Some statements, as the following one "However, we find that as far as solving the fundamental allocation problem is concerned here, the “real costs” of holding and the total cost are irrelevant" are very progressive without any arguments supporting it. 

Thank you for your hints, so I have changed it a bit. (r377-378)(r645)

Explain what kind of restrictions you mean, when you write “there are many restrictions” … when you use backwards running, recursive dynamic programming method, “in finding the solution” 

Like this: - start the stock with zero, stop the time horizon with zero, we need to know the exact setup costs an holding costs, the minimum order quantity must be fixed or almost fix.

Conclusions

In the last section of the paper, the conclusion needs to include the paper theoretical contribution, practical/managerial contribution, limitations and future research perspectives. Please add some more details such as the limitations, which would further enhance the professional quality of the article.

Thank you for your advice, so I changed it. (see r622-630)(r638-643)

The paper has good potential but at this stage the paper further minor changes and some more clarifications is needed on the issues presented.

---

## [Decision Letter · Decision Letter 1]

3 Jun 2021

Introducing the EPP house (topological space) method to solve MRP problems

PONE-D-21-09900R1

Dear Dr. Vasa,

We’re pleased to inform you that your manuscript has been judged scientifically suitable for publication and will be formally accepted for publication once it meets all outstanding technical requirements.

Kind regards,

Dragan Pamucar

Academic Editor

PLOS ONE

Additional Editor Comments (optional):

Reviewers' comments:

Reviewer's Responses to Questions

**Comments to the Author**

1. If the authors have adequately addressed your comments raised in a previous round of review and you feel that this manuscript is now acceptable for publication, you may indicate that here to bypass the “Comments to the Author” section, enter your conflict of interest statement in the “Confidential to Editor” section, and submit your "Accept" recommendation.

Reviewer #1: All comments have been addressed

Reviewer #2: All comments have been addressed

Reviewer #3: All comments have been addressed

2. Is the manuscript technically sound, and do the data support the conclusions?

Reviewer #1: Yes

Reviewer #2: Yes

Reviewer #3: Yes

3. Has the statistical analysis been performed appropriately and rigorously? 

Reviewer #1: Yes

Reviewer #2: Yes

Reviewer #3: Yes

4. Have the authors made all data underlying the findings in their manuscript fully available?

Reviewer #1: Yes

Reviewer #2: Yes

Reviewer #3: Yes

5. Is the manuscript presented in an intelligible fashion and written in standard English?

Reviewer #1: Yes

Reviewer #2: Yes

Reviewer #3: Yes

6. Review Comments to the Author

Reviewer #1: Recommended for publication. I will advice you to please read proof twice to remove language and content error if any.

Reviewer #2: The authors have made all changes in accordance with the recommendations.

I recommend for publication.

Reviewer #3: The manuscript satisfies the PLOS ONE criteria for publication. I accept all corrections and answers, therefore I suggest for accepting.

7. PLOS authors have the option to publish the peer review history of their article (what does this mean?). If published, this will include your full peer review and any attached files.

Reviewer #1: No

Reviewer #2: No

Reviewer #3: No

---

## [Editor Report · Acceptance letter]

10 Jun 2021

PONE-D-21-09900R1 

Introducing the EPP house (topological space) method to solve MRP problems 

Dear Dr. Vasa:

I'm pleased to inform you that your manuscript has been deemed suitable for publication in PLOS ONE. Congratulations! Your manuscript is now with our production department. 

Kind regards, 

on behalf of

Dr. Dragan Pamucar 

Academic Editor

PLOS ONE